

# Open Access uptake by universities worldwide

Nicolas Robinson-Garcia[1], Rodrigo Costas[2,3] and Thed N. van Leeuwen[2]

[1] Delft Institute of Applied Mathematics, Delft University of Technology, Delft, Netherlands
[2] Centre for Science and Technology Studies, Leiden University, Leiden, Netherlands
[3] Centre for Research on Evaluation, Science and Technology, University of Stellenbosch, Stellenbosch, South Africa

## ABSTRACT

The implementation of policies promoting the adoption of an open science (OS) culture must be accompanied by indicators that allow monitoring the uptake of such policies and their potential effects on research publishing and sharing practices. This study presents indicators of open access (OA) at the institutional level for universities worldwide. By combining data from Web of Science, Unpaywall and the Leiden Ranking disambiguation of institutions, we track OA coverage of universities' output for 963 institutions. This paper presents the methodological challenges, conceptual discrepancies and limitations and discusses further steps needed to move forward the discussion on fostering OA and OS practices and policies.

# INTRODUCTION

The implementation of policies promoting the adoption of an open science (OS) culture must be accompanied by indicators that allow monitoring the penetration of such policies and their potential effects on research publishing and sharing practices. In this paper we present open access (OA) indicators for universities worldwide. We analyse the presence of OA by type of access, field differences and comparisons with scientific impact and international collaboration. We explore discrepancies between the operationalization of OA indicators and the different definitions of OA provided in the literature.

The notion of OS goes back to the sixteenth century (*David, 2008*), but it has recently gained relevance as the European Union (EU) introduced it as a pivotal stone in their research programmes (*Moedas, 2015*). Within the different directives set up to achieve it, OA has become one of the first milestones. Initiatives such as Plan S (*Else, 2018a*; *Else, 2018b*) or the European Commission's Open Science Monitor (https://ec.europa.eu/info/research-and-innovation/strategy/goals-research-and-innovation-policy/open-science/open-science-monitor_en) exemplify such efforts and the prioritization of OA for these agencies, the latter being the tool the European Commission is using to monitor its uptake. However, more granular levels of analysis are needed to better understand how OA is expanding, which OA models are being implemented and what the potential side-effects of such models are. Universities have been

Corresponding author
Nicolas Robinson-Garcia,
n.robinsongarcia@tudelft.nl

supporting OA for many years now. The most common strategy has been to build and maintain institutional repositories, and introduce mandates that oblige their researchers to deposit pre- or postprints of their publications (*Harnad, 2007*; *Harnad et al., 2008*). There is also evidence of institutions promoting OA publications by sponsoring the costs derived from the article processing charges (APC) of open journals (*Gorraiz & Wieland, 2009*; *Gorraiz, Wieland & Gumpenberger, 2012*). In most cases, institutions are faced with the challenge of determining the success of such initiatives and monitoring the compliance of their researchers with international and national OA mandates. Initiatives such as the ranking of OA repositories (*Aguillo et al., 2010*) offer partial information on the share of OA availability at the institutional level, as they only provide details on the among of documents stored in institutional repositories, irrespective of the overall research output. Although valuable, it is still insufficient, as institutional repositories may not be the main vehicle used by researchers to make their outputs openly accessible (*Arlitsch & Grant, 2018*), and not all researchers, even at the same university, comply with their institutional mandates in the same manner.

Until recently, there were no more than estimates as to the amount of OA publications. However, the development of platforms like CrossRef, DOAJ or even Google Scholar, along with computational advancements on web scrapping, have led to a plethora of large-scale analyses to empirically identify OA literature (*Archambault et al., 2014*; *Van Leeuwen, Tatum & Wouters, 2018*; *Piwowar et al., 2018*; *Martín-Martín, Costas & Van Leeuwen, 2018b*). Overall, these studies report that around half of the scientific literature is freely available but point towards the increasing availability of publications which do not adhere strictly to what is considered OA. The game changer in this respect, has been Unpaywall (*Piwowar et al., 2018*), a product developed by the non-profit Our Research (https://ourresearch.org/), which tracks OA versions of published research with a document identifier (e.g., DOI), including e-prints self-archived in repositories, recently becoming the most standard mechanism to identify OA.

In this article we present a first attempt at analyzing OA at the institutional level. The main goal of the study is to provide methodological insights that can ease the analytical use and interpretation of OA indicators while providing a general overview of institutional OA uptake. Hence, the article is in nature descriptive, not aiming at responding specific research questions on OA uptake, but to set the basis so that more specific research questions can be developed in the future in an informed way. In order to achieve the proposed goal, we structure our findings in the following way. First, we inform on how OA is being achieved in different institutions and countries, describe national trends, and pathways by which OA is being expanded. Second, we deepen the analysis into green and gold OA types to understand potential discrepancies between the common understanding of these two types OA and how it is actually operationalized in Unpaywall. The results of this study have been recently incorporated to the 2019 edition of the Leiden Ranking released in May 2019 (*Van Leeuwen, Costas & Robinson-Garcia, 2019*) and a first version was presented at the ISSI 2019 Conference (*Robinson-Garcia, Costas & Van Leeuwen, 2019*).

[1]A detailed description the assignment of publications to fields is provided here https://www.leidenranking.com/information/fields It is important to note here that humanities journals publications are taken out of the Leiden Ranking publication set.

## MATERIALS & METHODS

In this article we use different sets of data sources and combine different methods to determine OA. Publication data is retrieved from the CWTS in-house version of the Web of Science. Unlike in the Leiden Ranking (restricted to article and reviews), here we report indicators for letters, articles and reviews (that is documents considered *citable*) indexed in the Science Citation Index Expanded, Social Sciences Citation Index and Arts & Humanities Citation Index for the 2014–2017 period. We link publications to the 963 universities identified in the Leiden Ranking database via their disambiguated list of institutional names, also hosted at CWTS (*Waltman et al., 2012*). Publications are assigned to five fields of science, following the methodology employed in the Leiden Ranking.[1] These fields are: Biomedical and Health Sciences, Life and Earth Sciences, Mathematics and Computer Science, Physical Sciences & Engineering, and Social Sciences and Humanities. The supplementary data offers indicators aggregated at the institutional level as detailed record metadata is subject to proprietary rights and cannot be openly shared.

For each publication, we identify if they are openly accessible and the type of OA by querying the Unpaywall information. A dump version of the Unpaywall database retrieved in April, 2019. Unpaywall relies on Digital Object Identifiers (DOI), which means that we will only include records which have a DOI assigned to them. Furthermore, the Unpaywall API does not label types of OA but records different pieces of evidence of OA availability of each publication. More information on the Unpaywall approach to OA is available at their User Guide offered for researchers (http://unpaywall.org/data-format).

Four types of OA are considered. These four types of OA are defined as follows:

- **Green OA.** Self-archived versions of a manuscript. Here the responsibility lies on the authors of the publication, or institutional colleagues such as central library staff members, who oversee depositing the document in a repository. This version of the document may not correspond with the final version of the publisher.
- **Gold OA.** This refers to journals which publish all their manuscripts in OA regardless of the business model they follow (e.g., publicly sponsored, author pays).
- **Hybrid OA**. Toll access (non-OA) journals make specific publications openly accessible usually after the author pays a fee, claiming an alleged need to account for potential losses derived from subscription fees.
- **Bronze OA.** This OA type was first suggested by *Piwowar et al. (2018)* and refers to free-to-read articles made available by publishers, without an explicit mention to any OA license. This OA is not subjected to copyright conditions set to be defined as OA (i.e., they do not ensure perpetual free access).

The labelling of OA types is described in Fig. 1 and already highlights some of the difficulties raised when trying to define what is actually OA (*Torres-Salinas, Robinson-Garcia & Moed, 2019*). The Unpaywall database provides a set of different pieces of OA evidence for each DOI. For each piece of evidence, we study all the metadata labels referring to the OA status of the publication. Thus, when one piece of evidence suggests that a paper

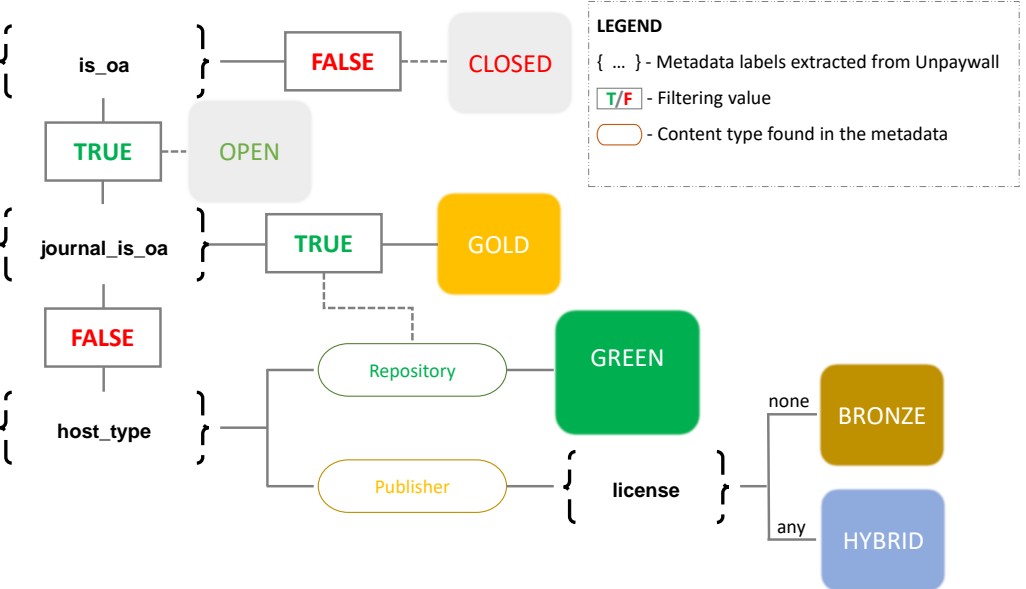

**Figure 1 Workflow followed to identify OA types based on Unpaywall data.** Source: *Van Leeuwen, Costas & Robinson-Garcia, 2019*.

belongs to an OA journal (gold OA), this automatically overrides bronze or hybrid OA. As observed in Fig. 1, gold, bronze and hybrid are conceptually incompatible types of OA, as these documents are all provided by the publishers, distinguishing themselves by the type of journal (OA or toll) or the presence of an OA license (hybrid or bronze). The only exception is made with green OA, which could overlap with any of the other three types.

Overall, a total of 4,621,721 distinct publications records are examined. 4,620,666 include a DOI, out of which 1,881,193 records were identified as OA. Figure 2 shows how these OA publications are distributed by type. 77% of all OA publications were green OA, followed by gold OA (33%), bronze OA (20%) and hybrid OA (16%). However, there is a substantial overlap between each of these latter OA types and green OA. 81% of all gold OA publications are also in green OA, for hybrid the share which is also green is 63%, and of hybrid are 45% for bronze OA.

The results are reported at different levels. First analyses investigate the share of OA on the overall output of each university, differences by country, continent and field. We then look specifically into the two main OA types: green and gold. We focus specifically on these two types as they represent the largest groups of OA publications. In the case of green OA publications, we focus on two aspects. First, the relation between green OA documents produced by an institution and share of which are stored in their institutional repository. Second, potential distortions on how OA types are operationalized and how they are commonly defined, specifically looking into the role played by PubMed Central (PMC). In the case of gold OA, we introduce different national models of gold OA publishing. We characterize gold OA publishing based on three variables: share of papers published in national journals, share of papers published in English language and share of papers

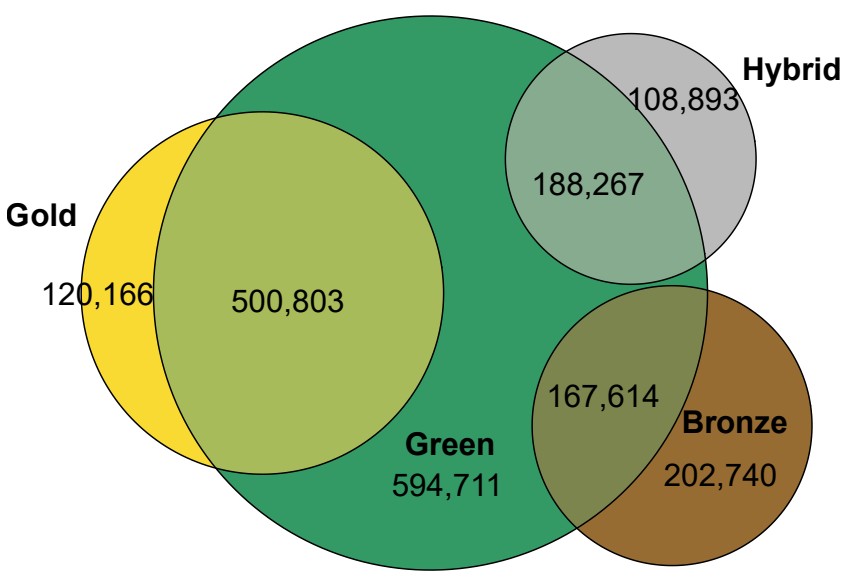

**Figure 2** Total number of documents in open access by type and overlap.

published in journals following including APCs. Language of documents and journal's country are identified using data from Web of Science. In the case of the latter, we identify the country of the journal by querying the field Publisher Address (PA). This approach is not exempt of limitations, as some publishers have an international outlook, while others may shape their geographic focus based on the location of their editors. Hence, the results of this analysis should be interpreted with caution and as a first attempt for developing taxonomies of gold OA publishing.

In the case of APCs, we queried the Directory of Open Access Journals (DOAJ). Here we must note that this is not a comprehensive list of OA journals. Unpaywall identifies a larger number of gold OA journals ($n = 11,601$) than DOAJ ($n = 11,365$), and for which we have no information on APCs. Therefore, the numbers on gold OA journals with/out APCs provided represent a lower bound of all the gold OA journals for which APC information is available via DOAJ. A total of 768 APC journals were identified. After some inspection, we found some inconsistencies in the way APC is defined according to DOAJ. That is, not in all cases, APC refers to an author pays model, but in some cases, journals offer an optional subscription fee for those interested on accessing to printed versions of the journal. This is the case for many journals stored in the SciELO platform which are free of costs for both readers and authors, but which offers the option to pay a subscription fee for printed versions of the journal.

## RESULTS

### General overview

In Fig. 3 we consider the proportion of OA publications by countries. Only countries with at least 10 universities listed in the Leiden Ranking are shown. The median share of publications openly available of universities worldwide is 43%. British universities have by

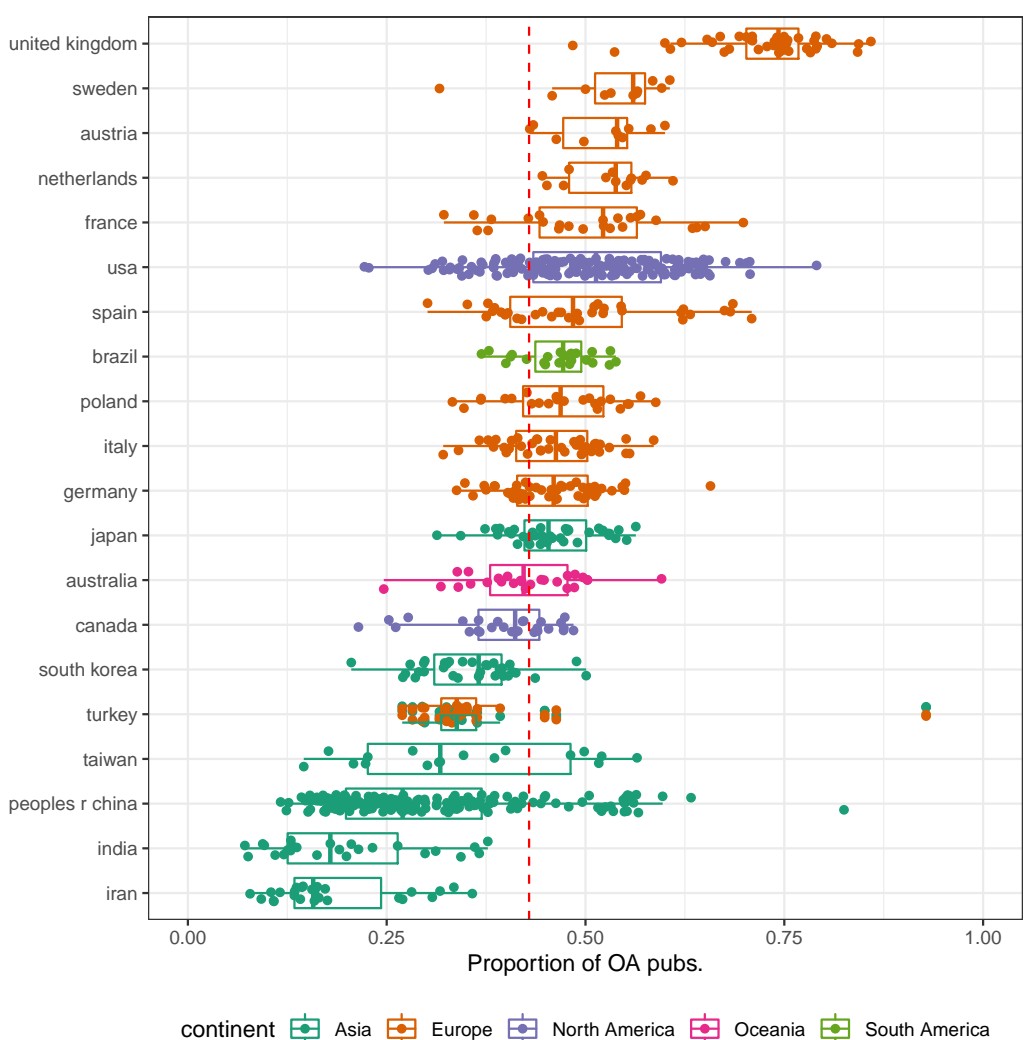

**Figure 3 Proportion of OA publications of the set of universities analysed by countries.** Only countries with at least 10 universities included are shown. Countries are ordered based on the median value of the share of OA publications of their universities. The red dashed line indicates the world median value. Turkey is assigned to both, Europe and Asia.

far the largest share of OA publications (median = 74%), followed by Sweden (median = 56%) and Austria (median = 54%). Except for the United States (median = 51%) and Brazil (median = 47%), all countries above world median are European. Asian countries, as well as Canada and Australia show OA shares below the world median.

We disaggregate by type of OA in Fig. 4. Most OA publications are openly accessible via the green route, and hence the similarity between Fig. 3 and Fig. 4A. In the case of gold OA (Fig. 4B) a very different image is seen. Brazilian universities stand out with a median of 30% publications in Gold OA. Sweden is placed in second, along with Taiwan (median = 18% for both countries). Universities from United Kingdom (median = 17%), Austria (median = 15%) and the Netherlands (median = 13%) correspondingly, show

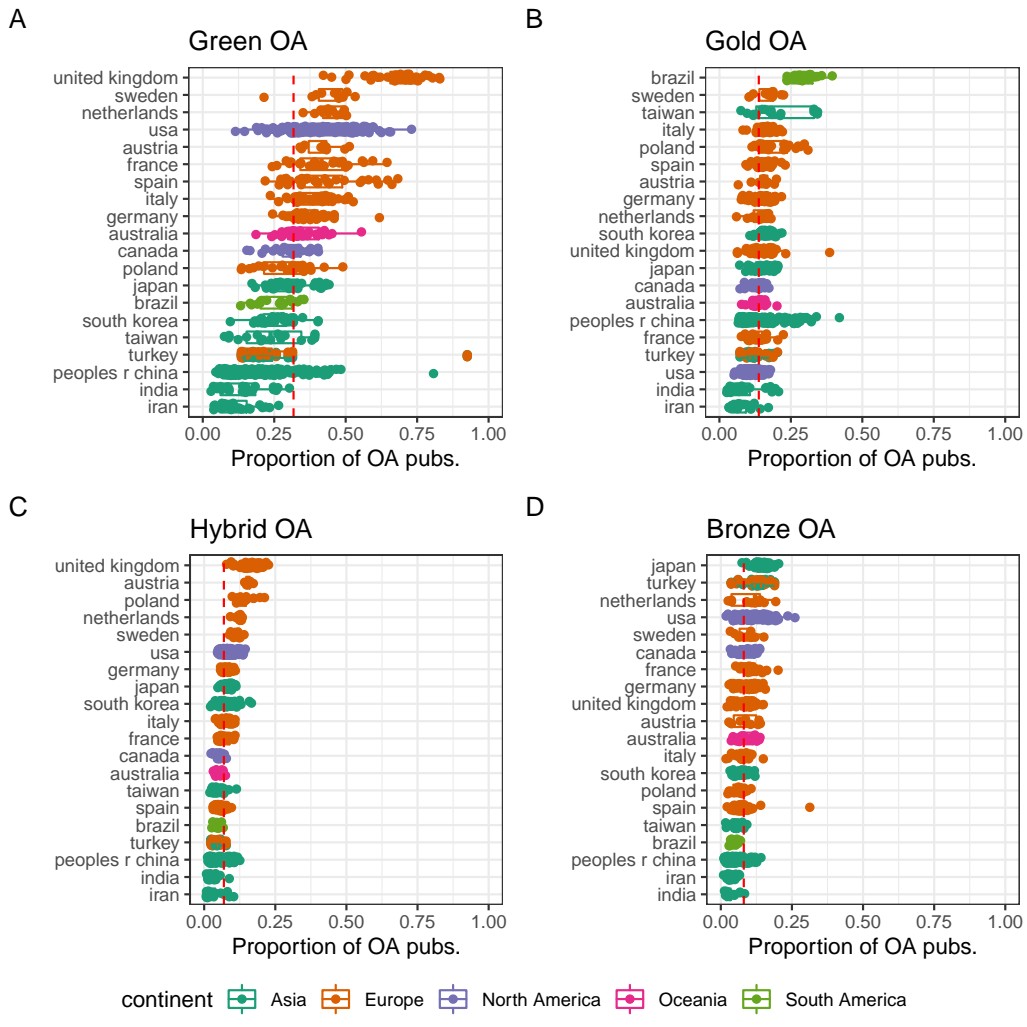

**Figure 4** **Proportion of OA publications of the set of universities analysed for A green OA, B gold OA, C hybrid OA and D bronze OA.** Only countries with at least 10 universities included are shown. Countries are ordered based on the median value of the share of OA p.

the highest share of hybrid OA publications. While for bronze OA, it is universities from Japan (median = 15%), Turkey (median = 13%) and the Netherlands (median = 12%) that stand out.

Figure 5 shows the predominance of each OA type by field and at the university level (each point represents the share of a university in each field and type of OA grouping). The average share of OA publications is 42.8%. The highest median is found in the Biomedical and Health Sciences (49.1%), while Social sciences and Humanities exhibit the lowest shares of OA (36.5%). Green OA is the most predominant form of OA regardless of the field (median of 33.2% in the 'All sciences' group). Again, the largest average is found in Biomedical and Health Sciences (39.0%) and the lowest in Social sciences and Humanities (28.0%). Overall, universities publish on average 14.7% of their publications in OA journals. For Biomedical & Health Sciences the average increases up to 19.3%, while in Mathematics

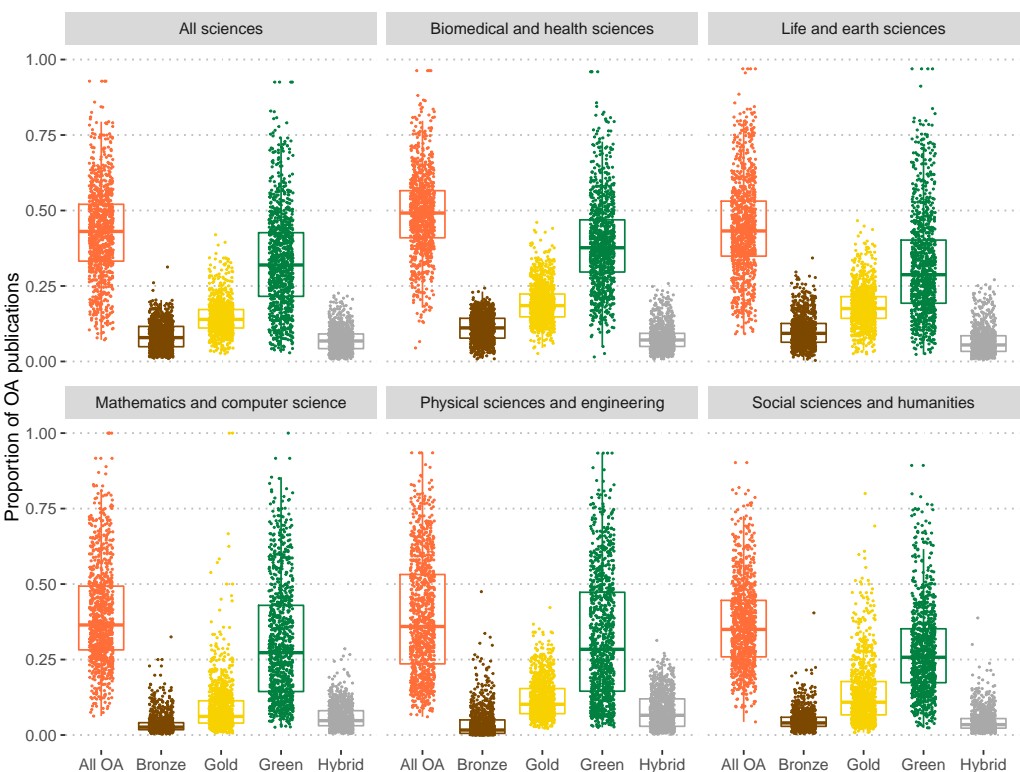

**Figure 5** **Proportion of OA publications of universities for each type of OA and for all OA types by field for universities worldwide.** Colors refer to each of the OA types. Orange: All OA; brown: bronze OA; yellow: gold OA; green: green OA; grey: hybrid OA.

& Computer Science it drops to 9.0%. In the case of Hybrid OA, an average of 7.1% of papers in universities are published under this modality. This figure increases in the case of Physical Sciences and Engineering to 7.9%, while in Social sciences & Humanities it represents an average of 4.6% of the output. Bronze OA, although it not strictly OA as it does not ensure sustainable access, is more common on average than Hybrid OA, with an overall share of 8.5% which goes up to 11.1% for Biomedical & Health Sciences, but with a presence on average of 3.7% in Mathematics & Computer Science.

We also note large differences by geographical region (Fig. 6). Europe (50.1%) and North America (49.1%) are the continents with the universities sharing the largest proportions of their output in OA. In the other extreme we find Asia (32.5%) and Africa (39.1%). In the former two continents, green OA is by large the most common OA type (41.1% in Europe and 40.6% in North America) with gold OA lagging behind by far as the second option (15.4% and 12.0% respectively). In South America, median shares of green (29.2%) and gold OA (27.0%) by university are practically identical. Shares of hybrid and bronze OA are on median below 10% for all continents except for bronze OA in North America (11.2%).

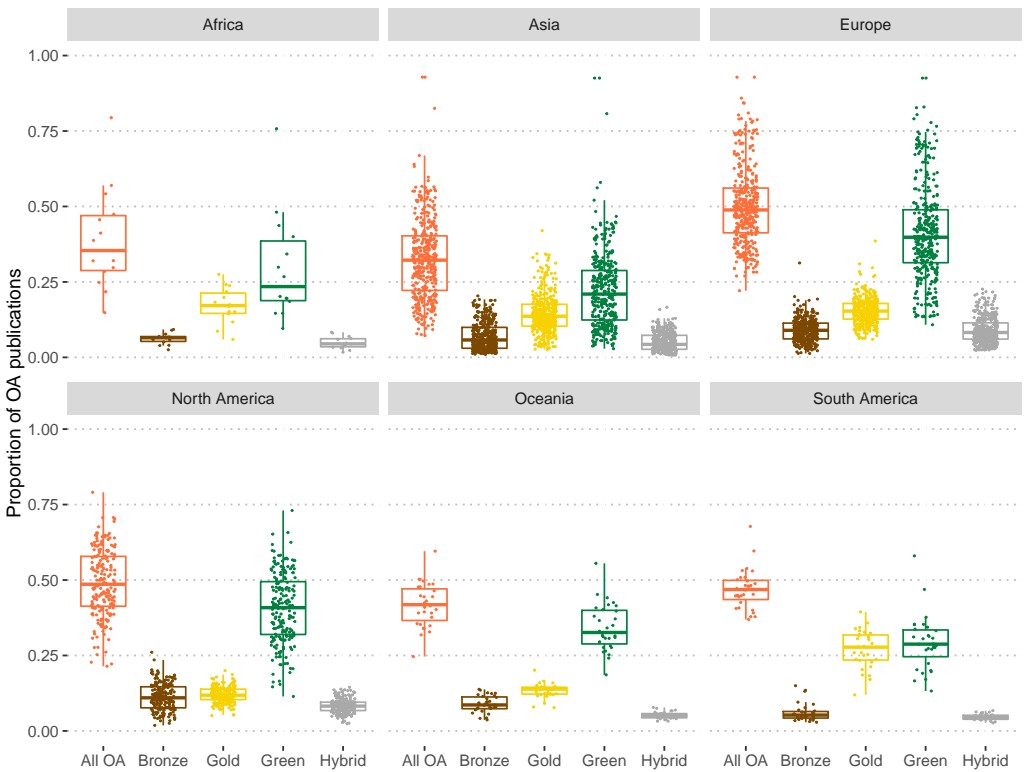

**Figure 6** **Proportion of OA publications of universities for each type of OA and for all OA types by region for universities worldwide.** Colors refer to each of the OA types. Orange: All OA; brown: bronze OA; yellow: gold OA; green: green OA; grey: hybrid OA.

## University profiling

It is remarkable that differences between and within universities can be quite significant. In Fig. 7 we take a closer look into the disciplinary profile of a set of universities based on the type and proportion of OA output by field. To illustrate the OA institutional profiling of universities, we use radar charts and select in each row the three universities with the largest output (considering their full counting) in North America, Europe, Africa, South America and Asia, respectively. In the first row, we observe the three largest universities in North America, two from the United States and one from Canada. The two US universities have above half of their output in green OA, with Social sciences and humanities, just below the 50% threshold. In the case of the University of Toronto, the shares are much lower, ranging between 39% green OA in Biomedical and Health Sciences and 3% bronze OA in Mathematics and Computer Science. The three largest universities in Europe are all from the United Kingdom. Again, green OA is clearly the most common OA option in all fields within these universities, showing more homogeneity across the three institutional profiles. However, Social Sciences and Humanities tend to have lower shares for the universities of Cambridge and Oxford than for University College London.

Regarding Africa (third row), two of the three universities showcased are South African, while the third one is Egyptian. In the case of Cairo University, no OA type in any field

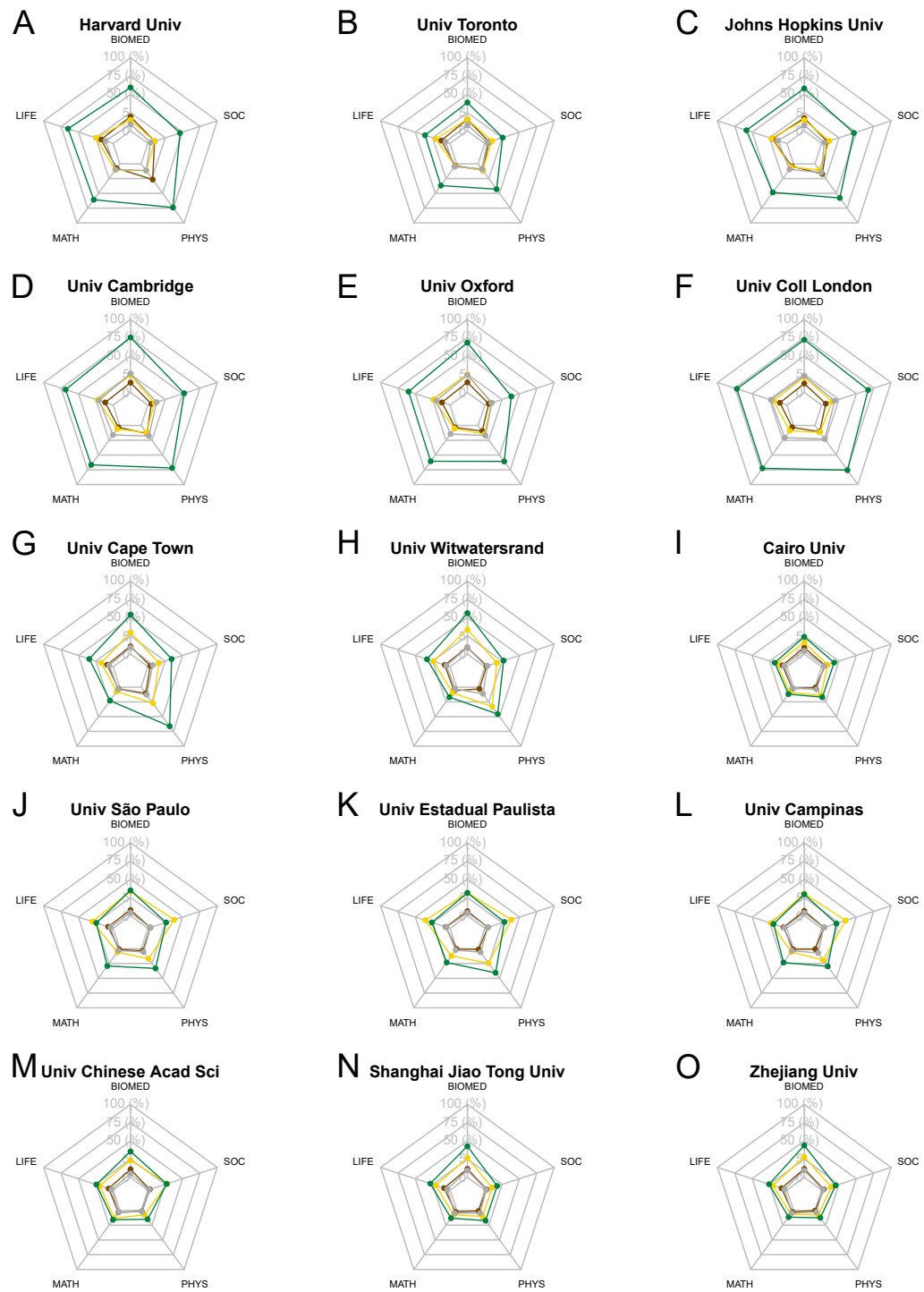

**Figure 7 An example of OA disciplinary profiles for top three universities with the largest output for North America (A–C), Europe (D–F), Africa (G–I), South America (J–L) and Asia (M–O).** Colors refer to each of the OA types. Brown: bronze OA; yellow: gold OA; green: green OA; grey: hybrid OA.

reaches a quarter of the total output of the university. In the other two cases, the profiles are quite similar, with the University of Cape Town exhibiting higher shares of green OA than the University of Witwatersrand. For South America, three Brazilian universities outstand as the largest ones; Universidade de São Paulo, Universidade Estadual Paulista and Universidade Estadual de Campinas. The gold OA preponderance previously observed at an aggregate level both for the continent and Brazil, is also noted at the institutional level in all three universities. However, we do observe that such preponderance is coming mainly from the Life and Earth Sciences and the Social Sciences and Humanities. Finally, for Asia (last row), we profile three Chinese universities for which green and gold OA shares go hand in hand in all three cases, with the exception of the field of Biomedical and Health Sciences, where green OA reaches higher shares of the total output.

## Green Open access and self-archiving

We will now delve into green OA, to better comprehend the indicators shown displayed on this typology. Green OA was originally defined as self-archiving of preprint or post print versions of published manuscripts. That means that green OA is achieved as the result of a proactive attitude of the authors or an institutional colleague, like librarians, towards OA. In their seminal paper, *Harnad et al. (2004)* go beyond such definition, and indicate that "the self-archiving method with the greatest potential to provide OA is self-archiving in the author's own university's OAI-compliant Eprint Archives" (p. 312). Hence, one could expect to see in the green OA indicators, shares of institutional self-archiving of a university's output. However, a closer look into what is considered as green following the identification procedure used based on Unpaywall data, shows that this is not the case for two reasons.

First, the assignment of OA output to each university is given based on the affiliation of authors and not the contents of institutional repositories. This means that universities with large proportions of their output in green OA may not be succeeding on storing their output in their institutional repositories themselves. Table 1 shows the top 20 universities with the largest shares of their output in green OA. Along with the total number of publications and green OA publications, we provide a threshold of the share of publications which are stored in their own institutional repository. We identify the lower band of the threshold by individually querying the URL string of each university's repository. The upper band results from also including URL string containing hdl.handle.net, which is the URL used when linking through the HANDLE identifier, a similar identifier to DOIs but assigned by repositories. Some universities do have most of their output accessible thanks to their own institutional repositories. For instance, 98% of green OA publications from Bilkent University are stored in their own repository.

Second, low coverages of green OA output in institutional repositories can be due to inter-institutional collaboration (i.e., collaboration with other institutional partners that apply more systematic archiving policies) or self-archiving in thematic (e.g., ArXiv) or supranational repositories (e.g., Zenodo). However, there is a third phenomenon which drifts further away the original definition of green OA from the actual numbers that are reported based on the general labelling obtained via Unpaywall. That is, the effect

**Table 1** **Top 20 universities with the highest share of their output available through green OA.** URL strings used are available in Supplemental Information 1.

| University | Country | Pubs | Green pubs | % Green pubs in Repository[a] |
|---|---|---|---|---|
| Bilkent Univ | Turkey | 2,008 | 1,858 | 97.7–97.7 |
| City Univ London | United Kingdom | 2,569 | 2,131 | 88.3–88.6 |
| Durham Univ | United Kingdom | 7,452 | 6,159 | 84.9–85.1 |
| Hong Kong Polytech Univ | China | 9,816 | 7,925 | 5.1–96.2 |
| London Sch Hyg | United Kingdom | 7,237 | 5,817 | 76.2–76.7 |
| Univ Strathclyde | United Kingdom | 4,847 | 3,830 | 88.9–89.0 |
| Univ St Andrews | United Kingdom | 5,780 | 4,497 | 79.2–79.7 |
| Loughborough Univ | United Kingdom | 4,274 | 3,271 | 85.9–86.2 |
| Univ Pretoria | South Africa | 6,432 | 4,873 | 93.7–93.7 |
| Univ Leeds | United Kingdom | 11,948 | 8,994 | 82.0–82.3 |
| Univ Glasgow | United Kingdom | 12,024 | 8,975 | 77.9–78.3 |
| Univ Bath | United Kingdom | 5,142 | 3,808 | 68.3–68.9 |
| Univ Edinburgh | United Kingdom | 18,139 | 13,415 | 55.2–58.2 |
| Caltech | United States | 13,481 | 9,834 | 69.2–69.4 |
| Univ Bristol | United Kingdom | 14,297 | 10,418 | 62.3–62.8 |
| Univ Reading | United Kingdom | 4,720 | 3,408 | 84.7–84.9 |
| London Sch Econ | United Kingdom | 3,525 | 2,534 | 79.4–79.83 |
| Univ Coll London | United Kingdom | 35,352 | 25,366 | 62.2–62.6 |
| Univ Sussex | United Kingdom | 5,510 | 3,931 | 69.1–69.7 |
| Univ Warwick | United Kingdom | 10,706 | 7,644 | 59.4–59.9 |

**Notes.**

[a]The interval refers to: lower bound when querying only for the institutional repository's URL string, and upper bound when querying for the institutional repository's URL string or *hdl.handle.net*.*When searching for the _hdl.handle.net_ string, the share increases to 73.8% of the total output.

of repositories which store OA documents without authors' intervention. Previously, we referred to this as different perspectives of green OA based on "the degree of engagement" of the authors (*Van Leeuwen, Costas & Robinson-Garcia, 2019*). We distinguish between two perspectives: (1) self-archiving, defined as the deliberate action of an author or librarian to archive publications in a repository, and (2) general archiving, where the archival function is still taking place, but without the explicit intervention of the author or librarian. So far, we have identified one macro repository following this general archiving perspective; PubMed Central (https://www.ncbi.nlm.nih.gov/pmc/). This source alone represents 60.8% of the green OA literature identified. However, some of its contents are retrieved from elsewhere, including OA journals such as Plos ONE. 86.5% of the 881,834 documents in PMC are simultaneously also gold, bronze or hybrid OA. The remaining 13.5% is accessible via another repository as well as PMC. As it is indeed a repository, in this study it is considered as a green OA source, but the effect of such decision in OA shares at the institutional level is highly significant. Figure 8 shows the effect of PMC on the shares of green OA. 49 universities are shown, these are those for which green OA deposited in PMC represents 95% or more of their total number of green OA publications. While self-archived and PMC publications can overlap (as more than one instance of OA evidence can be found
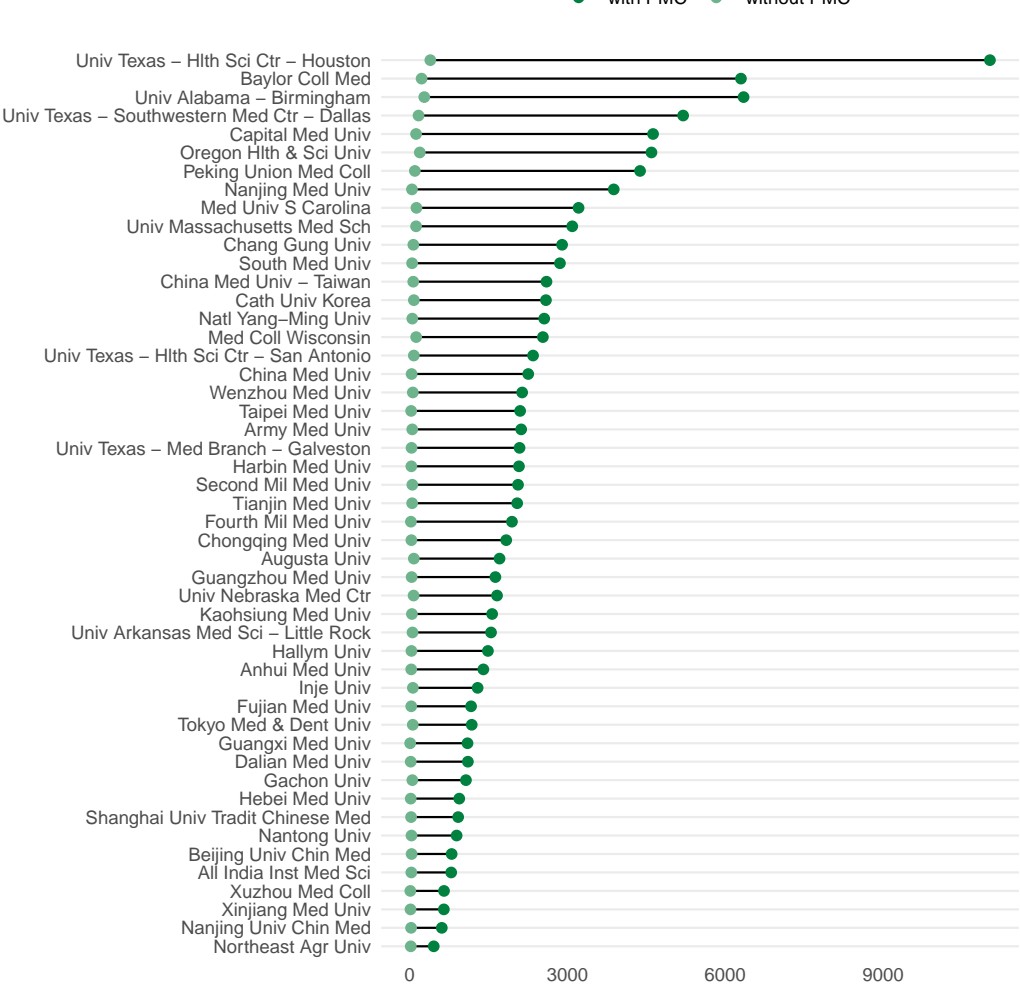

**Figure 8 Difference on number of green OA publications with and without PMC.** Only where PMC represents 95% or more of the share of green OA is shown.

per publication), in some cases the difference between defining PMC publications as green or not can derive on up to more than 10,000 publications, as in the case of University of Texas, Houston.

In Table 2 we aggregate the set of universities at the country level, to identify in which countries the inclusion of PMC as green OA affects the most their figures. The greatest effect is observed in Taiwan (65.5% of their total green OA), South Korea (62.2%) and China (60.1%). Furthermore, we observe that most of the documents coming from PMC are provided through another OA route, mostly gold, but also hybrid and bronze. This shows again the introduction of some degree of duplication of other OA types into green when including PMC and how the way we define and operationalize each of the OA types can affect the final numbers provided.

**Table 2  Top 20 countries with the highest share of distinct green OA publications coming from PMC.**

| Country | Green OA | PMC | PMC only | % Gold | % Bronze | % Hybrid |
|---|---|---|---|---|---|---|
| Taiwan | 18,841 | 14,748 | 12,337 | 5.59 | 0.52 | 0.75 |
| South Korea | 43,425 | 34,066 | 26,995 | 13.27 | 1.98 | 2.67 |
| China | 190,201 | 138,931 | 114,228 | 67.32 | 8.66 | 14.46 |
| Thailand | 5,166 | 3,987 | 2,578 | 61.05 | 11.14 | 12.47 |
| Lebanon | 819 | 620 | 386 | 61.77 | 10.97 | 11.13 |
| Egypt | 3,604 | 2,394 | 1,617 | 63.53 | 9.61 | 11.53 |
| Japan | 59,787 | 34,289 | 24,104 | 58.30 | 17.58 | 14.41 |
| Singapore | 10,717 | 6,637 | 4,266 | 61.22 | 13.56 | 12.88 |
| Malaysia | 8,675 | 4,718 | 3,345 | 81.37 | 4.60 | 7.31 |
| Poland | 19,672 | 10,222 | 7,404 | 56.54 | 5.34 | 29.94 |
| Pakistan | 1,344 | 638 | 496 | 80.41 | 3.76 | 9.09 |
| Austria | 18,208 | 10,554 | 6,471 | 45.26 | 10.79 | 31.20 |
| Canada | 71,913 | 45,445 | 25,121 | 51.15 | 16.98 | 11.55 |
| Iran | 8,412 | 4,408 | 2,931 | 47.84 | 5.24 | 26.66 |
| Brazil | 35,134 | 18,901 | 11,707 | 76.16 | 7.40 | 6.09 |
| India | 10,475 | 4,923 | 3,414 | 67.13 | 8.61 | 8.27 |
| USA | 522,934 | 383,483 | 169,403 | 30.14 | 17.96 | 11.11 |
| Israel | 16,761 | 8,750 | 5,407 | 51.77 | 16.32 | 13.46 |
| Mexico | 6,133 | 2,758 | 1,924 | 71.86 | 10.30 | 6.82 |
| Saudi Arabia | 10,042 | 5,211 | 3,108 | 71.73 | 7.29 | 9.56 |

## Gold Open access models

As previously observed, Gold OA is the second largest type of OA of the four analysed here (Fig. 2), but with some notable exceptions like the case of Brazil (Fig. 4B). *Torres-Salinas, Robinson-Garcia & Moed (2019)* highlight three models to characterize gold OA publishing from their analysis on Gold OA. The first one represents countries which publish in OA journals from big publishing firms and with a high Journal Impact Factor. Countries like United Kingdom, Germany or the Nordic countries fit into this model. A second model showcases countries publishing in national low Impact Factor OA journals, such as Brazil or India. The third model is a combination of the previous two, where they point out at countries like Poland or Spain. In Fig. 9 we take a similar approach looking at three variables for Gold OA publishing: share of gold OA publications in APC journals, share of gold OA publications in English language and share of gold OA publications from national journals. We observe that patterns are quite stable for the three variables. Most countries publish up to 25% of their output in national OA journals. APCs are paid for a range between 50% and 75% of their gold OA publications, and almost all of it is published in English language.

This pattern is followed by most countries, but some differences can be observed. For instance, United States and United Kingdom represent countries with high level of APC publishing, high shares in national language and almost exclusively in English language. Switzerland also fits into this pattern despite being a non-English speaking country. Another differing pattern is observed for countries like Spain or Portugal, where the share

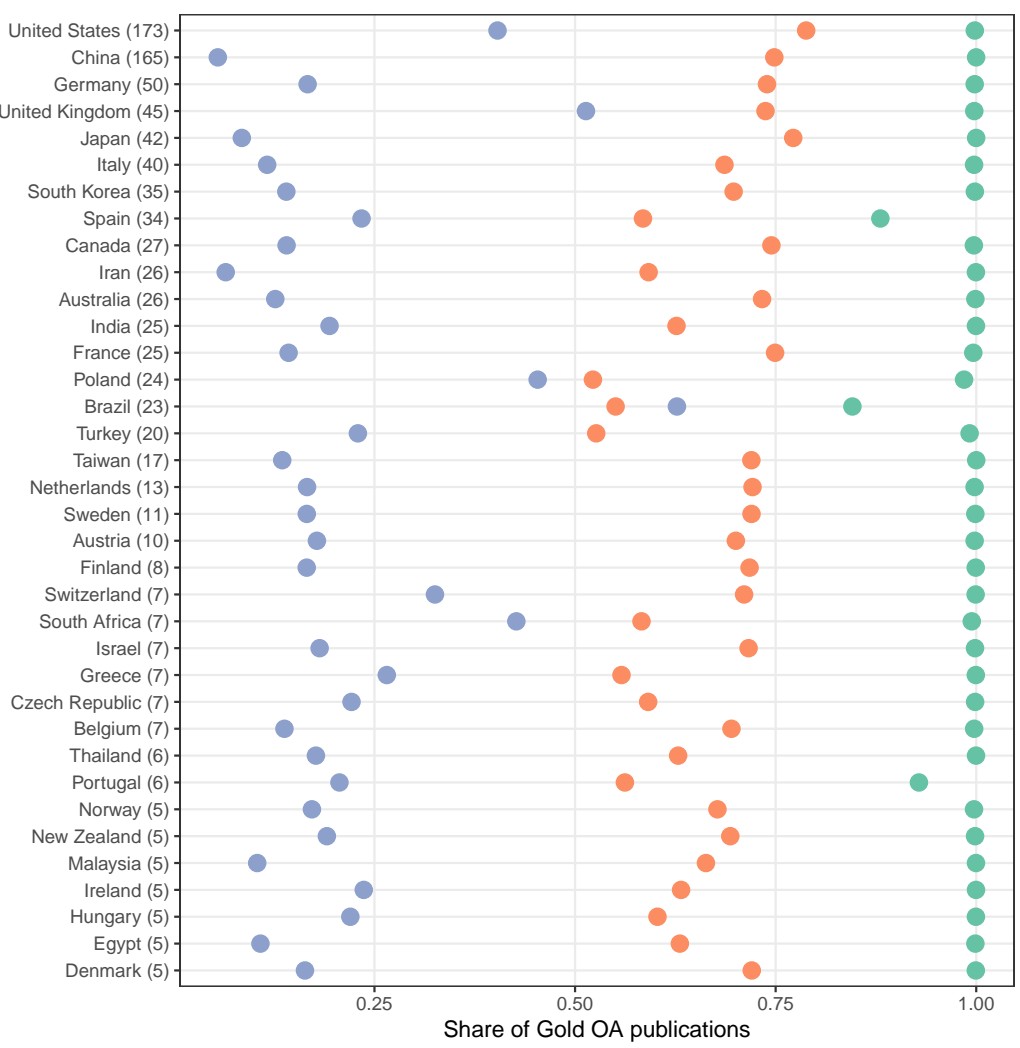

**Figure 9** **Share of Gold OA publications by country by type of publications.** Blue: Publications in national OA journals; Orange: Publications in APC OA journals; Green: Publications in English language. APC data is extracted from the Directory of Open Access Journals (DOAJ). Only countries with at least 5 universities in the Leiden Ranking are shown.

of English language publications is much lower although the share of national publications is still below 25% (23% and 21% respectively). In the case of Poland, although 98% of Gold OA publications are in English language, 45% come from national journals, with APC publications in the lower bound of the 50%–75% interval (52%). A similar pattern is followed by South Africa. Finally, we highlight the case of Brazil, where national gold OA publications represent 63% of the total of gold OA publications.

## DISCUSSION

The purpose of this study is to present a global view of the state of OA uptake at the institutional level. For this, we have included all universities appearing in the 2019th

edition of the Leiden Ranking and retrieved all their publications from Web of Science. These have been crossed with Unpaywall, a database which identifies evidences of OA for publications under the requirement that they have a DOI assigned to them. An important limitation of this tool is that it is dependent on DOIs, which means that we underestimate OA penetration overall, and especially in the Arts and Humanities fields (*Gorraiz et al., 2016*). Based on evidences of OA presence, we classified OA publications into four types: gold, green, hybrid and bronze. Overall, we find that around 41% of all publications contained in our data set are openly accessible. Green OA is the most common type of OA (77%), followed by Gold OA (33%).

Still, we find great differences between countries. For instance, Brazilian universities show a higher median share of Gold OA than Green OA, being the only case where this happen. This is a paradigmatic case, arguably the result of a long-standing OA policy commitment promoting national OA journals via the SciELO programme (*Meneghini, Mugnaini & Packer, 2006*). Furthermore, it goes beyond Brazilian universities and includes other South American countries such as Colombia or Chile (*Packer, 2009*; *Minniti, Santoro & Belli, 2018*), not necessarily fully covered in this study, due to the restrictions on the set of institutions included (only those present in the Leiden Ranking). United Kingdom, Netherlands, Austria and Sweden show similar levels of gold and hybrid OA, a surprising pattern as the levels of OA awareness and the types of mandates implemented in these countries is quite different (*Schmidt & Kuchma, 2012*). United Kingdom exhibit a strong OA uptake as a result of the implementation of policies which reinforce especially, the green route (*Chan, 2019*). In this sense, the UK's Research Evaluation Framework plays an important role on promoting OA, as any publication submitted for evaluation are required be openly accessible (*Hatzipanagos & Gregson, 2014*). These differences between countries are observed also at the continental level (Fig. 6) with Europe leading on OA penetration, followed by North America, and Asia and Africa lagging behind. However, it also yields many differences between universities from the same region, with only universities from Oceania and South America showing similar ratios of OA presence.

A closer look into green OA reveals some counterintuitive findings. First, the presence of repositories such as PubMed Central (PMC) which, although laudable, distort to some extent our perception of what is green OA and what it is not, particularly at the institutional level. This repository (and there might be others), indexes automatically OA literature, meaning that it includes self-archived publications as well as those from OA journals and OA publications from toll journals (Hybrid OA). Depending on how restrictive we are on our definition of green OA (i.e., self-archived by the author), we might disregard this source and hence reduce the overall presence of this type of OA. This along with the inclusion of bronze OA, evidence some discrepancies between the conceptual definition of OA and how it is operationalized in practice, leading the way to alternative conceptual framings of OA which might be closer to actual evidence of OA (e.g., *Martín-Martín et al., 2018a*). Here, we propose looking into the share of publications stored in universities' own repository and highlight some cases of good practices such as Bilkent University or City University London (Table 1).

In the case of gold OA, where the definition is much clearer, the intrusion of an author pays model (or APC model), along with the emergence of predatory journals (*Grudniewicz et al., 2019*), has led the way to much criticism as to the quality of OA journals (*Bohannon, 2013*). In the present study, the study of the presence of predatory journals in the counts of OA publications at the institutional-level is not directly approached, and we estimate that the study of this aspect may be restricted by the fact that we are only considering publications covered in the Web of Science database (from which we normalize institutional names and extract scientific fields), which is a database with a more restrictive coverage of scientific publications. But a similar approach to the one presented here could be performed to larger databases (such as Dimensions or Microsoft Academic Graph) in which predatory journals could have a stronger presence (an aspect which remains to a large extent unclear, and requires additional research).

While it is out of the scope of this study to analyse or compare the quality of OA journals, we do attempt to characterize such journals. For this, we expand on the modelling proposed by *Torres-Salinas, Robinson-Garcia & Moed (2019)*, and use three variables to characterize countries' gold OA publishing: language of publication, journals' editing country and the inclusion of an APC model (Fig. 9). This way we can identify outliers following alternative models of publishing (such as the aforementioned case of Brazil), evidencing that in some cases, publishing in OA journals is more related with other factors, such as publishing in national journals or non-English language rather than with the fact that the journal is offered in OA.

## CONCLUSION

While the study is descriptive in nature, it opens the opportunity for institutions, funding agencies and national science policy officers to better understand the expansion of OA in their country and better design and model effectives mandates of OA. Furthermore, new indicators can be designed which may fit into indicator frameworks of OS (*Schomberg et al., 2019*), moving away from metrics of excellence to metrics of openness and transparency.

## ACKNOWLEDGEMENTS

The authors would like to thank Henri de Winter for technical support, Jason Priem and Heather Piwowar, developers of Unpaywall, for fruitful discussions on the identification of OA types, and David Moher and Stefanie Haustein who reviewed this manuscript and improved it with their suggestions and comments. The opinions expressed in this study reflect only the author's view. The European Commission is not responsible for any use that may be made of the information it contains

### Funding

Nicolas Robinson-Garcia has received funding from the European Union's Horizon 2020 research and innovation programme under the Marie Sklodowska-Curie grant agreement

No 707404. Rodrigo Costas was funded by the South African DST-NRF Centre of Excellence in Scientometrics and Science, Technology and Innovation Policy (SciSTIP). There was no additional external funding received for this study. The funders had no role in study design, data collection and analysis, decision to publish, or preparation of the manuscript.

### Grant Disclosures

The following grant information was disclosed by the authors:
European Union's Horizon 2020 Research and Innovation: 707404.
South African DST-NRF Centre of Excellence in Scientometrics and Science, Technology and Innovation Policy (SciSTIP).

### Competing Interests

Rodrigo Costas and Thed N. van Leeuwen are involved on the development of the Open Science Monitor maintained by the European Commission.

### Author Contributions

- Nicolas Robinson-Garcia conceived and designed the experiments, performed the experiments, analyzed the data, prepared figures and/or tables, authored or reviewed drafts of the paper, and approved the final draft.
- Rodrigo Costas and Thed N. van Leeuwen conceived and designed the experiments, performed the experiments, authored or reviewed drafts of the paper, and approved the final draft.

### Data Availability

    The data is available in the Supplementary File.

### Supplemental Information

Supplemental information for this article can be found online at http://dx.doi.org/10.7717/peerj.9410#supplemental-information.

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
