# Peer review of "Open Access uptake by universities worldwide"

_PeerJ, doi:10.7717/peerj.9410_

## Round 0.1 · original submission · Major Revisions

One reviewer has suggested major revisions and has also provided an annotated manuscript. The other has suggested minor revisions. Please address all the comments in a detailed response with your resubmission.

·

Basic reporting

Please see general comments to authors.

Experimental design

Please see general comments to authors.

Validity of the findings

Please see general comments to authors.

Additional comments

The authors provide an examination of institutional penetration, globally, of Open Access (OA). Using the Leiden rankings and including 963 institutions the authors track OA penetration through use of the Unpaywall API – for records having a DOI. This methodological paper provides some preliminary results – OA penetration is slightly above 40%, globally, but there are wide disparities. For example, the UK does far better than other countries, such as Canada.

I found the methods and results transparently reported.

A major thought (perhaps concern as well) is the ‘basic’ manner in which the data were dealt with – descriptively. I was wondering whether some modelling (e.g., regressions) would make the results stronger. For example, do different types of OA and region of the world better predict institutional OA? I’m not used to thinking about PubMed Central (PMC) as an OA repository and wondered whether some modelling with and without PMC would be more informative (Figure 8 provides some sense of the disparity)? Similarly, given the overlap of types of OA, how did the authors handle the ‘counting’? were overlaps counted twice? Would regressions better model the different types of OA without overlap?

Some aspects of the results are not clearly explained. For example, why is the UK substantially better than other countries, such as India (Figure 3)?

The authors allude to the usefulness of these data to institutions and I agree with their assumption. What is less clear is how many institutions have the infrastructure in place to compute what the authors have done. I doubt many and wonder whether the authors should provide an example of how to work out the data for a particular institution; perhaps Delft? I think this worked example would be helpful to institutions wanting to enable this process (and better disseminating this paper).

Predatory journals advertise themselves as OA. Would the authors approach enable institutions get a better handle on which of their constituents (staff; faculty) are publishing in predatory journals?

·

Basic reporting

The article is very timely and provides a great way to integrate data sources such as the CWTS institutional disambiguation (from the Leiden ranking) and Unpaywall data to analyze OA uptake at universities worlwide.

The article needs quite a bit of improvement in the way that it is written and a thorough proofread. I have therefore made comments and edits in the attached PDF.

Experimental design

This is a great study analyzing the uptake of various OA types on the level of universities, disciplines and countries. It combines the method by Piwowar et al (2018, published in PeerJ) with data from the Leiden ranking.

A specific research question is lacking and the objective could be better defined. The research definitely fills a knowledge gap as OA uptake has not yet been investigated on the institutional level to such an extent and with the new Unpaywall dataset.

Methods would benefit from a bit more detail: when was data retrieved from Unpaywall, what publication years does the set span etc.

Validity of the findings

The findings are very interesting and based on valid methods. However, I am missing context in terms of previous findings or national policies etc.

Data is made available on the level of presented data, that is universities, countries etc. However, the underlying data based on the article level is not provided. I assume this is due to restrictions by proprietary data providers such as WoS and CWTS. I therefore was not able to validate the findings beyond the provided Excel file.

Authors should clarify what data they could and could not provide.

Additional comments

I would suggest that you focus on improving the writing and the objectives of the study. The data is excellent, but the way in which it is presented, could be much improved with relatively little work, particularly in terms of the writing.

I really like the way that the data was visualized, particularly the combination of bar charts and scatterplots as well as Figure 1. Could you clarify what software you used to produce figures?

There are some inconsistencies between the numbers and figures in the text as specified in the comments in the PDF. These need to be fixed.

I would also suggest that you provide percentages for OA types in the tables for countries and universities to facilitate comparison.

---

## Round 0.2 · accepted · Accept

Both referees report that you have met all of their concerns and so I am happy to accept your article for publication.

·

Basic reporting

This is revision. I have no further comments.

Experimental design

This is revision. I have no further comments.

Validity of the findings

This is revision. I have no further comments.

Additional comments

This is revision. I have no further comments. The authors have addressed my concerns in a reasonable manner.

·

Basic reporting

The authors have addressed all major concerns and improved the writing of the manuscript.

Experimental design

I disagree with the authors that a descriptive research article does not need research questions or objectives, but I think that the main research objective added is sufficient to clarify the goals of the paper.

Validity of the findings

no comment

Additional comments

Congratulations on a great and timely study!